# Masked Label Prediction: Unified Message Passing Model for Semi-Supervised Classification

## Abstract

Graph neural network (GNN) and label propagation algorithm (LPA) are both message passing algorithms, which have achieved superior performance in semi-supervised classification. GNN performs *feature propagation* by a neural network to make predictions, while LPA uses *label propagation* across graph adjacency matrix to get results. However, there is still no good way to combine these two kinds of algorithms. In this paper, we proposed a new **Uni**fied **M**essage **P**assaging Model (UniMP) that can incorporate *feature propagation* and *label propagation* with a shared message passing network, providing a better performance in semi-supervised classification. First, we adopt a Graph Transformer jointly label embedding to propagate both the feature and label information. Second, to train UniMP without overfitting in self-loop label information, we propose a masked label prediction strategy, in which some percentage of training labels are simply masked at random, and then predicted. UniMP conceptually unifies feature propagation and label propagation and be empirically powerful. It obtains new state-of-the-art semi-supervised classification results in Open Graph Benchmark (OGB).

## 1 Introduction

There are various scenarios in the world, e.g., recommending related news and products, discovering new drugs, or predicting social relations, which can be described as graph structures. Many methods have been proposed to optimize these graph-based problems and achieved significant success in many related domains such as predicting the properties of nodes (Yang et al., 2016; Kipf & Welling, 2016), links (Grover & Leskovec, 2016; Battaglia et al., 2018), and graphs (Duvenaud et al., 2015; Niepert et al., 2016; Bojchevski et al., 2018).

In the task of semi-supervised node classification, we are required to learn with labeled examples and then make predictions for those unlabeled ones. To better classify the nodes' labels in the graph, based on the Laplacian smoothing assumption (Li et al., 2018; Xu et al., 2018b), the message passing models were proposed to aggregate the information from its connected neighbors in the graph, acquiring enough facts to produce a more robust prediction for unlabeled nodes. Generally, there are two kinds of practical methods to implement message passing model, the Graph Neural Networks (GNNs) (Kipf & Welling, 2016; Hamilton et al., 2017; Xu et al., 2018b; Liao et al., 2019; Xu et al., 2018a; Qu et al., 2019) and the Label Propagation Algorithms (LPAs) (Zhu, 2005; Zhu et al., 2003; Zhang & Lee, 2007; Wang & Zhang, 2007; Karasuyama & Mamitsuka, 2013; Gong et al., 2016; Liu et al., 2019). GNNs combine graph structures by propagating and aggregating nodes features through several neural layers, which get predictions from *feature propagation*. While LPAs make predictions for unlabeled instances by *label propagation* iteratively.

Since GNN and LPA are based on the same assumption, making semi-supervised classifications by information propagation, there is an intuition that incorporating them together for boosting performance. Some superior studies have proposed their graph models based on it. For example, APPNP (Klicpera et al., 2019) and TPN (Liu et al., 2019) integrate GNN and LPA by concatenating them together, and GCN-LPA (Wang & Leskovec, 2019) uses LPA to regularize their GCN model. How-

ever, as shown in Tabel 1, aforementioned methods still can not incorporate GNN and LPA within a message passing model, *propagating feature* and *label* in both training and prediction procedure.

Table 1: Comparision between message passing models

| | Training | | Prediction | |
|---|---|---|---|---|
| | **Feature** | **Label** | **Feature** | **Label** |
| LPA | | ✓ | | ✓ |
| GCN | ✓ | | ✓ | |
| APPNP | ✓ | | ✓ | |
| GCN-LPA | ✓ | ✓ | ✓ | |
| UniMP (Ours) | ✓ | ✓ | ✓ | ✓ |

To unify the *feature* and *label propagation*, there are mainly two issues needed to be addressed:

**Aggregating feature and label information.** Since node feature is represented by embeddings, while node label is a one-hot vector. They are not in the same vector space. In addition, there are different between their message passing ways, GNNs can propagate the information by diverse neural structures like GraphSAGE (Hamilton et al., 2017), GCN (Kipf & Welling, 2016) and GAT (Veličković et al., 2017). But LPAs can only pass the label message by graph adjacency matrix.

**Supervised training.** Supervised training a model with *feature* and *label propagation* will overfit in self-loop label information inevitably, which makes the label leakage in training time and causes poor performance in prediction.

In this work, inspired by several advantages developments (Vaswani et al., 2017; Wang et al., 2018; Devlin et al., 2018) in Natural Language Processing (NLP), we propose a new **Uni**fied **M**essage **P**assing model (UniMP) with masked label prediction that can settle the aforementioned issues. UniMP is a multi-layer Graph Transformer, jointly using label embedding to transform nodes labels into the same vector space as nodes features. It propagates nodes features like the previous attention based GNNs (Veličković et al., 2017; Zhang et al., 2018). Meanwhile, its multi-head attentions are used as the transition matrix for propagating labels vectors. Therefore, each node can aggregate both features and labels information from its neighbors. To supervised training UniMP without overfitting in self-loop label information, we draw lessons from masked word prediction in BERT (Devlin et al., 2018) and propose a masked label prediction strategy, which randomly masks some training instances' label embedding vectors and then predicts them. This training method perfectly simulates the procedure of transducing labels information from labeled to unlabeled examples in the graph.

We conduct experiments on three semi-supervised classification datasets in the Open Graph Benchmark (OGB), where our new methods achieve novel state-of-the-art results in all tasks, gaining $82.56\%$ ACC in *ogbn-products*, $86.42\%$ ROC-AUC in *ogbn-proteins* and $73.11\%$ ACC in *ogbn-arxiv*. We also conduct the ablation studies for the models with different inputs to prove the effectiveness of our unified method. In addition, we make the most thorough analysis of how the label propagation boosts our model's performance.

## 2 METHOD

We first introduce our notation about graph. We denote a graph as $G = (V, E)$, where $V$ denotes the nodes in the graph with $|V| = n$ and $E$ denotes edges with $|E| = m$. The nodes are described by the feature matrix $X \in \mathbb{R}^{n \times f}$, which usually are dense vectors with $f$ dimension, and the target class matrix $Y \in \mathbb{R}^{n \times c}$, with the number of classes $c$. The adjacency matrix $A = [a_{i,j}] \in \mathbb{R}^{n \times n}$ is used to describe graph $G$, and the diagonal degree matrix is denoted by $D = \text{diag}(d_1, d_2, ..., d_n)$, where $d_i = \sum_j a_{ij}$ is the degree of node $i$. A normalized adjacency matrix is defined as $D^{-1}A$ or $D^{-\frac{1}{2}} A D^{-\frac{1}{2}}$, and we adopt the first definition in this paper.

### 2.1 FEATURE PROPAGATION AND LABEL PROPAGATION

In semi-supervised node classification, based on the Laplacian smoothing assumption, the GNN transforms and propagates nodes features $X$ across the graph by several layers, including linear layers and nonlinear activation to build the approximation of the mapping: $X \rightarrow Y$. The feature

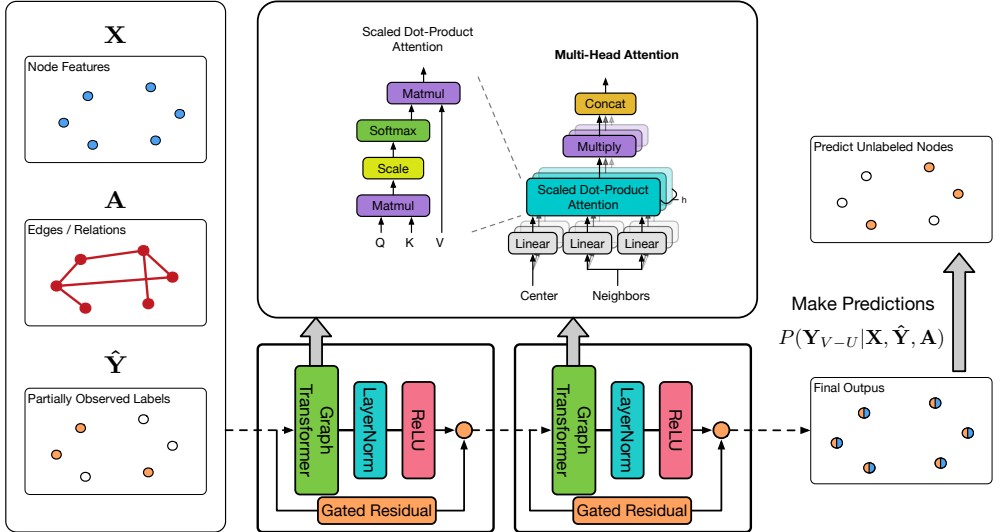

Figure 1: The architecture of our UniMP.

propagation scheme of GNN in layer $l$ is:

$$H^{(l+1)} = \sigma(D^{-1}AH^{(l)}W^{(l)})$$
$$Y = f_{out}(H^{(L)})$$

(1)

where the $\sigma$ is an activation function, $W^{(l)}$ is the trainable weight in the $l$-th layer, and the $H^{(l)}$ is the $l$-th layer representations of nodes. $H^{(0)}$ is equal to node input features $X$. Finally, a $f_{out}$ output layer is applied on the final representation to make prediction for $Y$.

As for LPA, it also assumes the labels between connected nodes are smoothing and propagates the labels iteratively across the graph. Given an initial label matrix $\hat{Y}^{(0)}$, which consists of one-hot label indicator vectors $\hat{y}_i^0$ for the labeled nodes or zeros vectors for the unlabeled. A simple iteration equation of LPA is formulated as following:

$$\hat{Y}^{(l+1)} = D^{-1}A\hat{Y}^{(l)}$$

(2)

Labels are propagated from each other nodes through a normalized adjacency matrix $D^{-1}A$.

## 2.2 UNIFIED MESSAGE PASSING MODEL

As shown in Figure 1, we employ a Graph Transformer, jointly using label embedding to construct our unified message passing model for combining the aforementioned feature and label propagation together.

**Graph Transformer**. Since Transformer (Vaswani et al., 2017) has been proved being powerful in NLP, we employ its vanilla multi-head attention into graph learning with taking into account the case of edge features. Specifically, given nodes features $H^{(l)} = \{h_1^{(l)}, h_2^{(l)}, ..., h_n^{(l)}\}$, we calculate multi-head attention for each edge from $j$ to $i$ as following:

$$q_{c,i}^{(l)} = W_{c,q}^{(l)}h_i^{(l)} + b_{c,q}^{(l)}$$
$$k_{c,j}^{(l)} = W_{c,k}^{(l)}h_j^{(l)} + b_{c,k}^{(l)}$$
$$e_{c,ij} = W_{c,e}e_{ij} + b_{c,e}$$
$$\alpha_{c,ij}^{(l)} = \frac{\langle q_{c,i}^{(l)}, k_{c,j}^{(l)} + e_{c,ij}\rangle}{\sum_{u\in\mathcal{N}(i)}\langle q_{c,i}^{(l)}, k_{c,u}^{(l)} + e_{c,iu}\rangle}$$

(3)

where $\langle q, k \rangle = \exp(\frac{q^T k}{\sqrt{d}})$ is exponential scale dot-product function and $d$ is the hidden size of each head. For the $c$-th head attention, we firstly transform the source feature $h_i^{(l)}$ and distant feature $h_j^{(l)}$ into query vector $q_{c,i}^{(l)} \in \mathbb{R}^d$ and key vector $k_{c,j}^{(l)} \in \mathbb{R}^d$ respectively using different trainable parameters $W_{c,q}^{(l)}, W_{c,k}^{(l)}, b_{c,q}^{(l)}, b_{c,k}^{(l)}$. The provided edge features $e_{ij}$ will be encoded and added into key vector as additional information for each layer.

After getting the graph multi-head attention, we make a message aggregation from the distant $j$ to the source $i$:

$$
\begin{aligned}
v_{c,j}^{(l)} &= W_{c,v}^{(l)} h_j^{(l)} + b_{c,v}^{(l)} \\
\hat{h}_i^{(l)} &= \Big\|_{c=1}^{C} \Big[ \sum_{j \in \mathcal{N}(i)} \alpha_{c,ij}^{(l)} (v_{c,j}^{(l)} + e_{c,ij}) \Big] \\
r_i^{(l)} &= W_r^{(l)} h_i^{(l)} + b_r^{(l)} \\
\beta_i^{(l)} &= \mathrm{sigmoid}(W_g^{(l)} [\hat{h}_i^{(l)}; r_i^{(l)}; \hat{h}_i^{(l)} - r_i^{(l)}]) \\
h_i^{(l+1)} &= \mathrm{ReLU}(\mathrm{LayerNorm}((1 - \beta_i^{(l)}) \hat{h}_i^{(l)} + \beta_i^{(l)} r_i^{(l)}))
\end{aligned}
\tag{4}
$$

where the $\|$ is the concatenation operation for $C$ head attention. Comparing with the Equation 1, multi-head attention matrix replaces the original normalized adjacency matrix as transition matrix for message passing. The distant feature $h_j$ is transformed to $v_{c,j} \in \mathbb{R}^d$ for weighted sum. In addition, inspired by (Li et al., 2019; Chen et al., 2020) to prevent oversmoothing, we propose a gated residual connections between layers by $r_i \in \mathbb{R}^d$ and $\beta_i^{(l)} \in \mathbb{R}^1$.

Specially, similar to GAT, if we apply the Graph Transformer on the output layer, we will employ averaging for multi-head output as following:

$$
\begin{aligned}
\hat{h}_i^{(l)} &= \frac{1}{C} \sum_{c=1}^{C} \Big[ \sum_{j \in \mathcal{N}(i)} \alpha_{c,ij}^{(l)} (v_{c,j}^{(l)} + e_{c,ij}^{(l)}) \Big] \\
h_i^{(l+1)} &= (1 - \beta_i^{(l)}) \hat{h}_i^{(l)} + \beta_i^{(l)} r_i^{(l)}
\end{aligned}
\tag{5}
$$

**Label Embedding and Propagation**. We propose to embed the partially observed labels information into the same space as nodes features: $\hat{Y} \in \mathbb{R}^{n \times c} \rightarrow \hat{Y}_e \in \mathbb{R}^{n \times f}$, which consist of the label embedding vector for labeled nodes and zeros vectors for the unlabeled. And then, we combine the label propagation into Graph Transformer by simply adding the nodes features and labels features together as propagation features $(H^0 = X + \hat{Y}_e) \in \mathbb{R}^{n \times f}$. We can prove that by mapping partially-labeled $\hat{Y}$ and nodes features $X$ into the same space and adding them up, our model is unifying both label propagation and feature propagation within a shared message passing framework. Let's take $\hat{Y}_e = \hat{Y} W_e$ and $A^*$ to be normalized adjacency matrix $D^{-1} A$ or the attention matrix from our Graph Transformer like Equation 3. Then we can find that:

$$
\begin{aligned}
H^{(0)} &= X + \hat{Y} W_c \\
H^{(l+1)} &= \sigma(((1 - \beta) A^* + \beta I) H^{(l)} W^{(l)})
\end{aligned}
\tag{6}
$$

where $\beta$ can be the gated function like Equation 4 or a pre-defined hyper-parameters like APPNP (Klicpera et al., 2019). For simplification, we let $\sigma$ function as identity function, then we can get:

$$
\begin{aligned}
H^{(l)} &= ((1 - \beta) A^* + \beta I)^l (X + \hat{Y} W_c) W^{(1)} W^{(2)} \dots W^{(l)} \\
&= ((1 - \beta) A^* + \beta I)^l X W + ((1 - \beta) A^* + \beta I)^l \hat{Y} W_c W
\end{aligned}
\tag{7}
$$

where $W = W^{(1)} W^{(2)} \dots W^{(l)}$. Then we can find that our model can be approximately decomposed into feature propagation $((1 - \beta) A^* + \beta I)^l X W$ and label propagation $((1 - \beta) A^* + \beta I)^l \hat{Y} W_c W$.

## 3 MASKED LABEL PREDICTION

Previous works on GNNs seldom consider using the partially observed labels $\hat{Y}$ in both training and inference stages. They only take those labels information as ground truth target to supervised train

their model's parameters $\theta$ with given $X$ and $A$:

$$\arg\max_{\theta} \quad \log p_\theta(\hat{Y}|X, A) = \sum_{i=1}^{\hat{V}} \log p_\theta(\hat{y}_i|X, A) \tag{8}$$

where $\hat{V}$ represents the partial nodes with labels. However, our UniMP model propagates nodes features and labels to make prediction: $p(y|X, \hat{Y}, A)$. Simply using above objective for our model will make the label leakage in the training stage, causing poor performance in inference. Learning from BERT, which masks input words and makes prediction for them to pretrain their model (masked word prediction), we propose a masked label prediction strategy to train our model. During training, at each step, we corrupt the $\hat{Y}$ into $\tilde{Y}$ by randomly masking a portion of node labels to zeros and keep the others remain, which is controlled by a hyper-parameter called label_rate. Let those masked labels be $\bar{Y}$, our objective function is to predict $\bar{Y}$ with given $X$, $\tilde{Y}$ and $A$:

$$\arg\max_{\theta} \quad \log p_\theta(\bar{Y}|X, \tilde{Y}, A) = \sum_{i=1}^{\bar{V}} \log p_\theta(\bar{y}_i|X, \tilde{Y}, A) \tag{9}$$

where $\bar{V}$ represents those nodes with masked labels. In this way, we can train our model without the leakage of self-loop labels information. And during inference, we will employ all $\hat{Y}$ as input labels to predict the remaining unlabeled nodes.

## 4 EXPERIMENTS

We propose a Unified Message Passing Model (UniMP) for semi-supervised node classification, which incorporates the feature and label propagation jointly by a Graph Transformer and employ a masked label prediction strategy to optimize it. We conduct the experiments on the Node Property Prediction of Open Graph Benchmark (OGBN), which includes several various challenging and large-scale datasets for semi-supervised classification, splitted in the procedure that closely matches the real-world application Hu et al. (2020). To verified our models effectiveness, we compare our model with others State-Of-The-Art models (SOTAs) in *ogbn-products*, *ogbn-proteins* and *ogbn-arxiv* three OGBN datasets. We also provide more experiments and comprehensive studies to show our motivation more intuitively, and how LPA improves our model to achieve better results.

### 4.1 DATASETS AND EXPERIMENTAL SETTINGS

Table 2: Dataset statistics of OGB node property prediction

| Name | Node | Edges | Tasks | Split Rate | Split Type | Task Type | Metric |
|------|------|-------|-------|-----------|-----------|-----------|--------|
| ogbn-products | $2,449,029$ | $61,859,140$ | 1 | $8\backslash02\backslash88$ | Sales rank | Multi-class class | Accuracy |
| ogbn-proteins | $132,534$ | $39,561,252$ | 112 | $65\backslash16\backslash19$ | Species | Binary class | ROC-AUC |
| ogbn-arxiv | $169,343$ | $1,166,243$ | 1 | $78\backslash08\backslash14$ | Time | Multi-class class | Accuracy |

**Datasets.** Most of the frequently-used graph datasets are extremely small compared to graphs found in real applications. And the performance of GNNs on these datasets is often unstable due to several issues including their small-scale nature, non-negligible duplication or leakage rates, unrealistic data splits (Dwivedi et al., 2020; Hu et al., 2020). Consequently, we conduct our experiments on the recently released datasets of Open Graph Benchmark (OGB) (Hu et al., 2020), which overcome the main drawbacks of commonly used datasets and thus are much more realistic and challenging. OGB datasets cover a variety of real-world applications and span several important domains ranging from social and information networks to biological networks, molecular graphs, and knowledge graphs. They also span a variety of predictions tasks at the level of nodes, graphs, and links/edges. As shown in table 2, in this work, we performed our experiments on the three OGBN datasets with different sizes and tasks for getting credible result, including *ogbn-products* about 47 products categories classification with given 100-dimensional nodes features , *ogbn-proteins* about 112 kinds of proteins function prediction with given 8-dimensional edges features and *ogbn-arxiv* about 40-class topics classification with given 128 dimension nodes features. More details about these datasets are provided in Appendix A.

**Implementation Details.** As mentioned above, these datasets are different from each other in sizes or tasks. So we evaluate our model on them with different sampling methods like previous studies (Li et al., 2020), getting credible comparison results. In *ogbn-products* dataset, we use Neighbor-Sampling with $size = 10$ for each layer to sample the subgraph during training and use full-batch for inference. In *ogbn-proteins* dataset, we use Random Partition to split the dense graph into subgraph to train and test our model. The number of partitions is 9 for training and 5 for test. As for small-size *ogbn-arxiv* dataset, we just apply full batch for both training and test. We set the hyper-parameter of our model for each dataset in Table 3, and the label_rate means the percentage of labels we preserve during applying masked label prediction strategy. We use Adam optimizer with $lr = 0.001$ to train our model. Specially, we set weight decay to $0.0005$ for our model in small-size *ogbn-arxiv* dataset to prevent overfitting. More details about the tuned hyper-parameters are provided in Appendix B.

Table 3: The hyper-paramerter setting of our model

|                 | ogbn-products    | ogbn-proteins    | ogbn-arxiv |
|-----------------|------------------|------------------|------------|
| sampling_method | NeighborSampling | Random Partition | Full-batch |
| num_layers      | 3                | 7                | 3          |
| hidden_size     | 128              | 64               | 128        |
| num_heads       | 4                | 4                | 2          |
| dropout         | 0.3              | 0.1              | 0.3        |
| lr              | 0.001            | 0.001            | 0.001      |
| weight_decay    | *                | *                | 0.0005     |
| label_rate      | 0.625            | 0.5              | 0.625      |

## 4.2 Comparison with SOTAs

Baseline and other comparative SOTAs are provided by OGB leaderboard. Some of the including results are conducted officially by authors from original papers, while the others are re-implemented by communities. And all these results are guaranteed to be reproducible with open source codes. Following the requirement of OGB, we run our experimental results for each dataset 10 times and report the mean and standard deviation. As shown in Tabel 4, Tabel 5, and Tabel 6, our unified model outperform all other comparative models in three OGBN datasets. Since most of the compared models only consider optimizing their models for the features propagation, these results demonstrate that incorporating label propagation into GNN models can bring significant improvements. Specifically, we gain 82.56% ACC in *ogbn-products*, 86.42% ROC-AUC in *ogbn-proteins*, which achieves about 0.6-1.6% absolute improvements compared to the newly SOTA methods like DeeperGCN (Li et al., 2020). In *ogbn-arxiv*, our method gains 73.11% ACC, achieve 0.37% absolute improvements compared to GCNII (Chen et al., 2020), whose parameters are four times larger than ours.

Table 4: Results for ogbn-products

| Model                           | Test Accuracy         | Validation Accuracy   | Params      |
|---------------------------------|-----------------------|-----------------------|-------------|
| GCN-Cluster (Chiang et al., 2019) | $0.7897 \pm 0.0036$ | $0.9212 \pm 0.0009$   | $206,895$   |
| GAT-Cluster                     | $0.7923 \pm 0.0078$   | $0.8985 \pm 0.0022$   | $1,540,848$ |
| GAT-NeighborSampling            | $0.7945 \pm 0.0059$   | -                     | $1,751,574$ |
| GraphSAINT (Zeng et al., 2019)  | $0.8027 \pm 0.0026$   | -                     | $331,661$   |
| DeeperGCN (Li et al., 2020)     | $0.8090 \pm 0.0020$   | $0.9238 \pm 0.0009$   | $253,743$   |
| **UniMP**                       | $\mathbf{0.8256 \pm 0.0031}$ | $\mathbf{0.9308 \pm 0.0017}$ | $1,475,605$ |

Table 5: Results for ogbn-proteins

| Model                           | Test ROC-AUC          | Validation ROC-AUC    | Params      |
|---------------------------------|-----------------------|-----------------------|-------------|
| GaAN (Zhang et al., 2018)       | $0.7803 \pm 0.0073$   | -                     | -           |
| GeniePath-BS (Liu et al., 2020b)| $0.7825 \pm 0.0035$   | -                     | $316,754$   |
| MWE-DGCN                        | $0.8436 \pm 0.0065$   | $0.8973 \pm 0.0057$   | $538,544$   |
| DeepGCN (Li et al., 2019)       | $0.8496 \pm 0.0028$   | $0.8921 \pm 0.0011$   | $2,374,456$ |
| DeeperGCN (Li et al., 2020)     | $0.8580 \pm 0.0017$   | $0.9106 \pm 0.0016$   | $2,374,568$ |
| **UniMP**                       | $\mathbf{0.8642 \pm 0.0008}$ | $\mathbf{0.9175 \pm 0.0007}$ | $1,909,104$ |

Table 6: Results for ogbn-arxiv

| Model | Test Accuracy | Validation Accuracy | Params |
|---|---|---|---|
| DeeperGCN (Li et al., 2020) | $0.7192 \pm 0.0016$ | $0.7262 \pm 0.0014$ | $1,471,506$ |
| GaAN (Zhang et al., 2018) | $0.7197 \pm 0.0024$ | - | $1,471,506$ |
| DAGNN (Liu et al., 2020a) | $0.7209 \pm 0.0025$ | - | $1,751,574$ |
| JKNet (Xu et al., 2018b) | $0.7219 \pm 0.0021$ | $0.7335 \pm 0.0007$ | $331,661$ |
| GCNII (Chen et al., 2020) | $0.7274 \pm 0.0016$ | - | $2,148,648$ |
| **UniMP** | $\mathbf{0.7311 \pm 0.0021}$ | $\mathbf{0.7450 \pm 0.0005}$ | $473,489$ |

### 4.3 ABLATION STUDIES ON GRAPH TRANSFORMER AND MASKED LABEL PREDICTION

In this section, we will conduct extensive studies to identify the improvements from different components of our unified model. To get a fair comparison, we re-implement classical GNN methods like GCN and GAT, following the same sampling methods and model setting shown in Table 3. The hidden size of GCN is head_num*hidden_size since it doesn't have head attention. We also change different inputs for our models to study the effectiveness of feature and label propagation.

As shown in Tabel 7, it's surprising that only $\hat{\mathbf{Y}}$ and $\mathbf{A}$, GNNs still work well in all three datasets, outperforming those MLP model only given $\mathbf{X}$. This implies that one's label relies heavily on its neighborhood instead of itself feature. For models with $\mathbf{X}$ and $\mathbf{A}$ as inputs like most GNNs do, they are more likely to remember the labels of training set through approximations, which is inaccurate. It's a waste of information in semi-supervised classification when prediction without incorporating the annotated label $\hat{\mathbf{Y}}$ information from training sets, which are preciser than the model's approximations for training data. In addition, with different input settings, our improved Graph Transformer can outperform GAT, GCN in most cases.

Table 7: Ablation studies on models with different inputs.

| Inputs | Model | Datasets | | |
|---|---|---|---|---|
| | | ogbn-products Test ACC | ogbn-proteins [†] Test ROC-AUC | ogbn-arxiv Test ACC |
| $\mathbf{X}$ | MLP | $0.6106 \pm 0.0008$ | $0.7204 \pm 0.0048$ | $0.5765 \pm 0.0012$ |
| $\mathbf{X, A}$ | GCN | $0.7851 \pm 0.0011$ | $0.8265 \pm 0.0008$ | $0.7218 \pm 0.0014$ |
| | GAT | $0.8002 \pm 0.0063$ | $0.8376 \pm 0.0007$ | $0.7246 \pm 0.0013$ |
| | Transformer | $0.8137 \pm 0.0047$ | $0.8347 \pm 0.0014$ | $0.7292 \pm 0.0010$ |
| $\mathbf{A}, \hat{\mathbf{Y}}$ | GCN | $0.7832 \pm 0.0013$ | $0.8083 \pm 0.0021$ | $0.7018 \pm 0.0009$ |
| | GAT | $0.7751 \pm 0.0054$ | $0.8247 \pm 0.0033$ | $0.7055 \pm 0.0012$ |
| | Transformer | $0.7987 \pm 0.0104$ | $0.8160 \pm 0.0007$ | $0.7090 \pm 0.0007$ |
| $\mathbf{X, A}, \hat{\mathbf{Y}}$ | GCN | $0.7987 \pm 0.0104$ | $0.8247 \pm 0.0032$ | $0.7264 \pm 0.0003$ |
| | GAT | $0.8193 \pm 0.0017$ | $0.8556 \pm 0.0009$ | $0.7278 \pm 0.0009$ |
| | Transformer | $\mathbf{0.8269 \pm 0.0009}$ | $\mathbf{0.8560 \pm 0.0003}$ | $\mathbf{0.7332 \pm 0.0014}$ |

[†] In *ogbn-proteins*, nodes features are not provided initially. We average the edge features as their nodes features and drop the edge features for fair comparison in this experiment. which is slightly different from Table 5. $\mathbf{X}$ is the nodes features, $\mathbf{A}$ is the graph adjacent matrix and $\hat{\mathbf{Y}}$ is the observed labels. We run these models three times and report their means and stds.

### 4.4 EXPLORING HOW LABEL PROPAGATION AFFECTS UNIMP

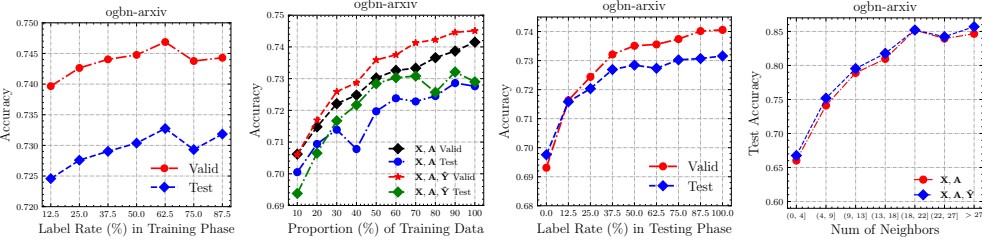

(a) Training with different label rate. (b) Training with different proportion of data. (c) Testing with different proportion of labels. (d) Test accuracy with different neighbors.

Figure 2: Exploration of how label coverage affects label propagation.

One of our motivations for using label propagation is that labels can carry additional informative feature which cannot be replaced by the model's approximation. However, the relation between the coverage of labeled data and the impact of label propagation for our model still be uncertain. Therefore, we conduct more experiments in *ogbn-arxiv* to investigate their relationship in several different scenarios:

- In Figure 2a, we train UniMP using $\mathbf{X}, \hat{\mathbf{Y}}, \mathbf{A}$ as inputs. We tune the training label rate which is the hyper-parameter of masked label prediction task and display the validation and test accuracy. Our model achieves better performance when label rate is about 0.625.

- Figure 2b describes the correlation between the proportion of training data and the effectiveness of label propagation. We fix the label rate with 0.625. The only change is the training data proportion. It's a common sense that with the increase of training data, the performance is gradually improving. And the model with label propagation can have greater benefits from increasing labeled data proportion.

- In the training stage, our model always masks a part of the training label and tries to recover them. But in the inference stage, our model utilizes all training labels for predictions, which is slightly inconsistent with the one in training. In Figure 2c, we fix our trained models and perform label propagation with different label rate in inference. It's found that when lower the label rate during prediction, UniMP might have worse performance (less than 0.70) than the baseline (about 0.72). However, when the label rate climbs up, the performance can boost up to 0.73.

- In Figure 2d, we calculate accuracy for unlabeled nodes with a different number of neighbors. The experimental result shows that nodes with more neighbors have higher accuracy. And the model with label propagation can always have improvements even with different numbers of training neighbors.

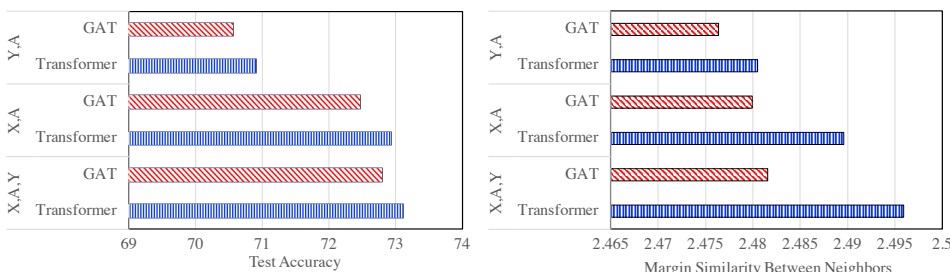

Figure 3: Correlation between accuracy and margin similarity between neighbors.

Wang & Leskovec (2019) have theoretically proved that using the LPA for GCN during training can enable nodes within the same class/label to connect more strongly, increasing the accuracy (ACC) of model's prediction. Our model can be seen as an upgraded version of them, using LPA in both training and testing time. Therefore, we try to experimentally prove the above idea based on our model. We use the Margin Similarity Function to reflect the connection tightness of the nodes with same class (the higher scores, the stronger connection they are, and more details in Appendix C). We conduct the experiments on *ogbn-arxiv*. And as shown in Figure 3, the ACC of models' prediction is proportional to Margin Similarity. Unifying feature and label propagation can further strengthen their connection, improving their ACC. Moreover, our Graph Transformer outperforms GAT in both connection tightness and ACC with different inputs.

## 5 CONCLUSION

We first propose a unified message passing model, UniMP, which jointly performs feature propagation and label propagation within a Graph Transformer to make the semi-supervised classification. Furthermore, we propose a masked label prediction method to supervised training our model, preventing it from overfitting in self-loop label information. Experimental results show that UniMP outperforms the previous state-of-the-art models on three main OGBN datasets: *ogbn-products*, *ogbn-proteins* and *ogbn-arxiv* by a large margin, and ablation studies demonstrate the effectiveness of unifying feature propagation and label propagation.

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

## A    DATASETS DETAILS

**ogbn-products.** As shown in Table 2, *ogb-products* is an undirected and unweighted graph, representing an Amazon product co-purchasing network. The goal of this task is to predict the category of a product in a multi-class classification setup, where the 47 top-level categories are used for target labels. To match the real-world application, it conducts a splitting on the dataset based on sales ranking, where the top 10% for training, next top 2% for validation, and the rest for testing.

**ogbn-proteins.** As shown in Table 2, *ogbn-proteins* dataset is an undirected, weighted, and typed (according to species) graph. Nodes represent proteins, and edges indicate different types of biologically meaningful associations between proteins, e.g., physical interactions, co-expression or homology. The task is to predict the presence of protein functions in a multi-label binary classification setup, where there are 112 kinds of labels to predict in total. The performance is measured by the average of ROC-AUC scores across the 112 tasks. It conducts the splitting by species.

**ogbn-arxiv.** As shown in Table 2, *ogbn-arxiv* dataset is a directed graph, representing the citation network between all Computer Science (CS) arXiv papers indexed by MAG (Wang et al., 2020). The task is to predict the 40 subject areas of arXiv CS papers, e.g., cs.AI, cs.LG, and cs.OS, which are manually determined by the paper's authors and arXiv moderators. This dataset were splitted by time.

## B    HYPER-PARAMETERS TUNED ON UNIMP MODEL

There are the hyper-parameters we tuned on our unified model for comparison with other SOTA results, where the asterisks denote the hyper-parameters we eventually selected.

Table 8: The tuned hyperparamerters of our model

|  | **ogbn-prdouct** | **ogbn-proteins** | **ogbn-arxiv** |
|---|---|---|---|
| sampling_method | NeighborSampling | Random Partition | Full-batch |
| num_layers | [3*, 4] | [3, 5, 7*, 9] | [3*, 4] |
| hidden_size | [128*, 256] | [32, 64*, 128] | [128*, 256] |
| num_heads | [4*,2] | [6, 4*, 2] | [2*, 1] |
| dropout | [0.3*] | [0, 0.1*, 0.3] | [0.1, 0.3*] |
| lr | [0.01, 0.001*] | [0.01, 0.001*] | [0.1, 0.001*] |
| weight_decay | - | - | [0, 0.0005*] |
| label_rate | [0.625*] | [0.375, 0.5*, 0.625] | [0.125, 0.25, 0.375, 0.5, 0.625*, 0.75, 0.875] |

## C    MARGIN SIMILARITY FUNCTION

Given an attention weight $\alpha_{ij}$ from GAT or Graph Transformer, which can represent the connection tightness between source node $i$ and distance node $j$, we employ the Circle Loss (Sun et al., 2020) and make a slight change on it to build our Margin Similarity Function (MSF), measuring the connection tightness between the neighbors nodes with same labels. For each center node $i$ and its neighbors $j, k \in \mathcal{N}(i)$, we take the measurement task as a pair similarity problem in which the center node's neighbors with same label are positive samples and the others are negative samples, calculating their connection tightness as following:

$$MSF = \frac{1}{N} \sum_{i=1}^{N} \log \left( 1 + \sum_{j \in \mathcal{N}(i)_{pos}, k \in \mathcal{N}(i)_{neg}} e^{\alpha_{i,j}} - e^{\alpha_{i,k}} \right) \tag{10}$$