# OpenReview forum: "Masked Label Prediction: Unified Message Passing Model for Semi-Supervised Classification"
_ICLR.cc/2021/Conference — Reject_

### Official Review · AnonReviewer2 · 2020-10-26
**Official Blind Review #2**

**Rating:** 7
**Confidence:** 4

**Review:**

Summary:

The paper proposes a new Graph Transformer (UniMP) based model with the motive of combining two powerful semi-supervised node classification techniques, GNN and LPA. Proposed Graph Transformer unifies feature and label propagation in conjunction to provide a better performance in semi-supervised node property classification task. UniMP also benefits from a masked label prediction strategy which is inspired by [Devlin et al.](https://arxiv.org/abs/1810.04805)
Proposed method is able to achieve SOTA performance on several datasets from Open Graph Benchmark.

------------------------------------------------------------------------------------------------------------------------------------------------------------------------------------

Pros:

+ The method proposed for unifying feature and label propagation is simple yet sufficiently novel.

+ This paper is able to support all the claims with rigorous experimentation. The experimentation is done using Open Graph Benchmark datasets and the proposed method achieves SOTA on all the datasets used.

+ The paper is very well written and is easy to understand. I personally like the simple yet effective nature of the method.

------------------------------------------------------------------------------------------------------------------------------------------------------------------------------------

Questions:

- How do you map label information to feature information space? Is it a linear model? Did you explore any other embedding approaches before deciding on a linear model?

------------------------------------------------------------------------------------------------------------------------------------------------------------------------------------

Overall, I vote for accepting the paper. This paper provides a novel method to unify GNN and LPA algorithms in order to take advantage of feature as well as label information. Proposed method is well supported by rigorous experimentation on various datasets and ablation studies on various inputs.

------------------------------------------------------------------------------------------------------------------------------------------------------------------------------------

Minor Comments/Typos:

- Section 1: In addition, there are **different** between → In addition, there are **differences** between

- Section 4: To **verified** our model → To **verify** our model

- Section 4.2: Some of the **including** results → Some of the **included** results

- Section 4.3: instead of **itself** **feature** → instead of **it's features**

- Section 4.4: model still **be** uncertain → model still **remains** uncertain

---

> ### Author Response · Authors · 2020-11-13
> **Response to Reviewer 2**
>
> We thank the reviewer for taking the time to thoroughly read and his/her positive comments on our manuscript.
>
> **Q1: How do you map label information to feature information space?**
>
> A1: We only try the linear model to map our label information to feature space, since it is the most intuitive way. We believe that our work can inspire new research ideas to explore a more interesting way to unify GNN and LPA algorithms in semi-supervised classification.

---

### Official Review · AnonReviewer4 · 2020-10-26
**More discussions could be added**

**Rating:** 6
**Confidence:** 4

**Review:**

This paper proposed a novel unified massage passing strategy for semi-supervised graph learning scenario. It combines the advantages of the GNN and label propagation algorithms.

In the model, the label information is projected into the feature space, then merged/added with the node feature for the GNN feature aggregation module. By this way, the label information could be more effectively utilized in the semi-supervised scenario. In addition, a masked label prediction strategy is proposed to reduce the negative influence of the label leakage problem. The extensive experimental results demonstrate the effectiveness of the proposed model.

Pros:

This paper proposed a novel framework to more effectively and explicitly utilize the label information by GNN in semi-supervised scenario, and the experiments illustrate its effectiveness in general benchmark datasets.

Cons:

The insight/motivation of this method is not very significant. Compared with other methods such as APPNP,  TPN and GCN-LPA, the major differences is another effective/unified network structure to merge feature and label information together. Compared with GCN-LPA, the improvements in the insight/motivation aspect is not very significant. This paper shows empirical discussion compared with GCN-LPA, while it could be better to show some insight/theoretical discussion/analysis.

The improvements compared with other baselines are not significant. If more parameters/weights are deployed in other baselines, could the performance be better? Please discuss this point.

---

> ### Author Response · Authors · 2020-11-13
> **Response to Reviewer 4**
>
> Really appreciate your recognition of our contributions and the questions your raised for our paper. Here we would like to provide more discussions on the comparison between our model with other unified network structures.
>
> **Q1: About the insight/motivation of UniMP.**
>
> A1: The most important motivation of our work is that labels of the neighborhood are vital information in graph neural network settings.
>
> In the traditional machine learning scenarios, where predictions for instances are independent, labels in the training sets can't be used during testing. However, in graph neural network settings, unlabeled nodes in the testing phase are still connected with the labeled one in training sets. Therefore, it gives a chance to utilize the training label to predict those unlabeled ones. And we can see a rising trend of graph neural network communities trying to find a better way to utilize label information in their models[A][B][C][D].
>
> However, Table 1 has pointed out that the existed unified methods like APPNP[A], TPN[B], and GCN-LPA[C] are incorporating label information in a more indirect way. For example, APPNP and TPN only use the nodes feature $X$ as input, predicts a soft label, and then propagate them. As for GCN-LPA, they take the nodes features $X$ and labels $\hat{Y}$ as input in training test, using the training labels to regularize their GAT attention weight between nodes, while their GAT model can only use $X$ in inference.
>
> All the above models can not propagate nodes features and labels in both training and test time, while our model first propose the masked labeled prediction method to address this issue, incorporating feature and label propagation within a message passing model.
> - The experiment in table 7 shows that our method can also be applied to improve other kinds of GNNs.
> - Moreover, our unified model can take the additional validation labels as input to further boost model's performance, while other GNNs or unified networks do not have this advantage (They just use the validation to select hyper-perparameters and early stop their training step). We also provide related experiments.
>
>   model &emsp;&emsp;&emsp;&emsp;&emsp;&emsp;&emsp;&emsp;&emsp;&emsp;&emsp;&emsp;&emsp;&emsp;&emsp;&emsp;&emsp;&emsp;&emsp;&emsp;&emsp;&emsp;&emsp;&emsp;&emsp;&emsp;&emsp;&emsp;&nbsp;&nbsp;| ogbn-arxiv(test)
>   Unimp w/o Masked label prediction &emsp;&emsp;&emsp;&emsp;&emsp;&emsp;&emsp;&emsp;&emsp;&emsp;&emsp;&nbsp;&nbsp;&nbsp;&emsp;&emsp;&emsp;| 0.7293 ± 0.0010
>   Unimp w/ Masked label prediction (training label) &emsp;&emsp;&emsp;&emsp;&emsp;&emsp;&nbsp;&emsp;&emsp;| 0.7332 ± 0.0014
>   Unimp w/ Masked label prediction (training label + validation label) | **0.7377 ± 0.0002**
>
> **Q2: Compared with GCN-LPA, the improvements in the insight/motivation aspect is not very significant.**
>
> A2: As mentioned above, GCN-LPA uses LPA to regularize GAT’s attention weight of connected edges to improve their performance. It makes a serious limitation that their method can only optimize attention base GNN. However, our masked label prediction strategy can be applied in many kinds of GNNs (GCN and GAT) to improve their performance, shown in table 7.
>
> **Q3: If more parameters/weights are deployed in other baselines, could the performance be better?**
>
> A3: We show our OGBN leaderboard results in Table 4\~6 with given models’ parameters. Our model is not the largest while gaining SOTA results in three datasets.
> Moreover, the ablation studies in table 7 also demonstrate that our masked label prediction strategy can incorporate node feature $X$ and $\hat{Y}$ in many kinds of GNNs. Therefore, we can combine any state-of-the-art GNNs with our strategy.
>
> [A] Johannes Klicpera, Aleksandar Bojchevski, and Stephan Gunnemann. Predict then propagate: Graph neural networks meet personalized pagerank. arXiv: Learning, 2019.
>
> [B] Yanbin Liu, Juho Lee, Minseop Park, Saehoon Kim, Eunho Yang, Sung Ju Hwang, and Yi Yang. Learning to propagate labels: Transductive propagation network for few-shot learning. arXiv: Learning, 2019.
>
> [C] Hongwei Wang and Jure Leskovec. Unifying graph convolutional neural networks and label propagation. arXiv: Learning, 2019.
>
> [D] Qu M, Bengio Y, Tang J. Gmnn: Graph markov neural networks[J]. arXiv preprint arXiv:1905.06214, 2019.

---

### Official Review · AnonReviewer3 · 2020-10-27

**Rating:** 4
**Confidence:** 4

**Review:**

The paper presents a novel unified model that jointly harnesses the power of graph convolutional networks and label propagation algorithms based on the unified message passing framework. The UniMP first employs graph Transformer networks to jointly propagate both feature and label information. Then, to avoid label leakage, a masked label prediction strategy is employed.

Pros:
* The presented method shows strong empirical performance on the open graph benchmark dataset.
* The whole framework is simple and the idea is easy to follow.

Cons:
* What is the label leakage problem? It is not clear to me (1) why label will be leaked during the joint learning process and (2) what the outcome does label leakage bring.
* The writing of this paper is poor. The authors are suggested to polish their paper. Please see minor comments below.
* Although the proposed UniMP achieves state-of-the-art performance, the experiments are not convincing enough.
  * Experimental results are not consistent, c.f. Table 4~6 and Table 7. It seems that the standalone Transformer even surpasses UniMP on ogbn-products and ogbn-arxiv.
  * It seems that the hyper-parameter specifications vary greatly across the three datasets. To me the residual connection is helpful when stacking many layers (Li et al., 2019; Chen et al., 2020), while in this paper the number of layers is relatively low. A sensitivity analysis on the network depth is necessary to demonstrate the impact of the residual connection.
  * When compared with GAT, the main differences are (1) different implementations of self-attention mechanisms and (2) whether to adopt gated residual connections. However, no ablation studies are provided to demonstrate the impact of these two independent components. Especially, as shown in Table 7 and Figure 3, given $X$,$A$,$\hat{Y}$, transformer outperforms GAT. The authors are expected to elaborate on which component (self-attention implementation or gated residual connection) brings the improvement.

Minor comments:
* Abstract: we adopt a Graph Transformer jointly [using] label embedding?
* Abstract: UniMP ... and be empirical powerful -> is empirical powerful
* Page 2: there are different -> they are different
* Mathematical notations are severely abused; for example, hidden_size should be represented by $f$, and how $\hat{Y}_e$ is transformed from $\hat{Y}$ is not clear.

---

> ### Author Response · Authors · 2020-11-13
> **Response to Reviewer 3**
>
> We really thank the reviewer for his/her careful reading of the manuscript and constructive remarks. Here, we would like to provide more explanations to address the reviewer’s concerns.
>
> **Q1: What is the label leakage problem?**
>
> A1: As our unified model will take the node feature $X$ and partially observed label $\hat{Y}$ as input, when being trained, the model will overfit in their self-loop node labels information, invalidating their estimated performance. More concretely, when training labels are considered as model inputs, the model will tend to just simply copy the input labels and learn nothing about the relation between nodes. The objective function is   $\arg\max_{\theta} \log p_{\theta}( \hat{Y} |  X, A, \hat{Y})$.
>
> However, in testing time, those unlabeled nodes only take their nodes features as input, using the neighborhood labels and nodes to make predictions as $p( Y_{unlabeled} |  X, A, \hat{Y})  $. It will be lead to poor performance.
>
> To avoid the above label leakage problem, we propose a masked label prediction (MLP) strategy to train our model. During training, we corrupt the $\hat{Y}$ into $\tilde{Y}$ by randomly masking a portion of node labels to zeros and keep the others remain. Let those masked labels be $\bar{Y}$, our new objective function is to predict $\bar{Y}$ with given $X$, $\tilde{Y}$ and $A$. The formula can be rewritten as $\arg\max_{\theta}  \log p_{\theta}(\bar{Y}| X,\tilde{Y}, A)$ with $\hat{Y} =  [\bar{Y}; \tilde{Y}]$.
>
> Here, we extend the ablation studies in ogbn-arxiv to support this.
>
> Model  &emsp;&emsp;&emsp;&emsp;&emsp;&emsp;&emsp;&emsp;&nbsp;&nbsp;&nbsp;| train | valid | test
> UniMP ($A, X$) &emsp;&emsp;&emsp;&emsp;&emsp;&nbsp;&nbsp;| 0.8070 | 0.7414 | 0.7292
> UniMP ($A, X, \hat{Y}$) w/ MLP &nbsp;| 0.8116 | **0.7462** | **0.7332**
> UniMP ($A, X, \hat{Y}$) w/o MLP|  **0.9999** | 0.6169 | 0.6084
>
> UniMP without MLP strategy can easily get nearly 100\% training accuracy but poor validation score and test score, which is the consequence of label leakage. Without MLP strategy, labels will do more harm than good to our models. In table 7, by default, we have used MLP strategy to train the model with $A, \hat{Y}$ and $A, X, \hat{Y}$.
>
> **Q2:  Experimental results are not consistent, c.f. Table 4\~6 and Table 7. Standalone Transformer even surpasses UniMP.**
>
> A2: Table 4\~6 is the submitted result for OGB leaderboard which is actually graph Transformer + MLP strategy. And it has the same architecture with row 10 table 7 with ($A, X, \hat{Y}$) as input. We can find that UniMP (Transformer + $A, X, \hat{Y}$ inputs + MLP) outperforms Transformer + ($A, X$) inputs.
>
> Since the numerical scores are slightly different between Table 4\~6 and Table 7, we want to make more clarifications:
> -  In Table 7, all the experiments are conducted 3 times, while the scores in Table 4~6 are reported as 10 times average following the rules of OGB leaderboard.
> - For ogbn-proteins in Table 7, we average the edge features as nodes features and drop edge features, for making a fair comparison between our model and others, because vanilla GCN and GAT are incapable of handling edge features. Therefore, without edge features, the UniMP of ogbn-proteins in Table 7 has a lower score than that in Table 5.
> - The above clarifications can be found in the footnote of Table 7.
>
> **Q3: About the hyper-parameter specifications**
>
> A3: We get the initial hyper-parameter from other baseline models [A][B] from the leaderboard, without over tuning it on the datasets. The tuned hyper-parameters can be found in Appendix B.
>
> **Q4: About the difference compared with GAT and whether adopting gated residual connections.**
>
> A4: We provide more ablation studies in ogbn-arxiv to make comparisons from the following 3 aspects: (1) Vanilla transformer (dot-product) and GAT (sum attention); (2) w/ and w/o residual; (3) residual and gated residual.
>
> Model &emsp;&emsp;&emsp;&emsp;&emsp;│Transformer&emsp;&nbsp;|&emsp;GAT
> ogb-arxiv dataset | valid | test &emsp;&emsp;| valid | test
> w/o residual &emsp;&emsp;&nbsp;| 0.7402 | 0.7293 | 0.7405 | 0.7278
> w/ residual &emsp;&emsp;&nbsp; &nbsp;&nbsp;| 0.7444 | 0.7311 | 0.7444 |0.7297
> w/ gated residual | 0.7462 | **0.7332** | 0.7435 | 0.7295
>
> We can find that dot-product attention can outperform sum attention. And gated residual can surpasses the one w or w/o simple residual in Transformer.
>
> Moreover, followed the official implement [A] in the leaderboard, GAT for ogb-product and ogb-proteins in Table 7 have been with residual connection.
>
> Model &emsp;&emsp;&emsp;&emsp;&emsp;&emsp;&emsp;&emsp;&emsp;│ ogbn-product (test) |  ogbn-porteins (test)
> GAT + residual&emsp;&emsp;&emsp;&emsp;&emsp;&emsp;&emsp;&emsp;| 0.8193 &emsp;| &emsp;0.8556
> transformer + gated residual &emsp;| **0.8269** &emsp;| &emsp;**0.8560**
>
> [A] https://github.com/rusty1s/pytorch_geometric/blob/master/examples/ogbn_products_gat.py
>
> [B] https://github.com/lightaime/deep_gcns_torch/tree/master/examples/ogb

---

### Official Review · AnonReviewer1 · 2020-10-28
**Review ---- Masked Label Prediction: Unified Message Passing Model for Semi-Supervised Classification**

**Rating:** 5
**Confidence:** 4

**Review:**

The authors proposed a unified message passing model to make a graph neural network to be able to incorporate both label propagation and feature propagation. Compared to previous work, the proposed model can also make use of partial label information in both training and inference stages. Experiments on three OGBN datasets show that the proposed methods achieve promising performance.

Pros:

    The motivation of the paper is very clearly stated in the text and the experiments successfully show that by incorporate label propagation and feature propagation, the performance can be improved.

    The partial label information is included in both training and inference stages.

Cons:

    In the literature review, one very important work [A], which is very closely related to the proposed model is ignored. In [A], a unified model for neural message passing is already established, and the proposed model seems to be a special case of the neural message passing module in [A]. The authors should compare their model to the one in [A] and clearly state the difference between the proposed model and the one [A], and should justify why should not use [A] directly.

[A] Gilmer, Justin, et al. "Neural Message Passing for Quantum Chemistry." ICML. 2017.

---

> ### Author Response · Authors · 2020-11-13
> **Response to Reviewer 1**
>
> We really thank the reviewer for his/her careful reading of the manuscript and constructive remarks, which helps improve the manuscript. Here, we would like to provide more explanations to address the reviewer's concerns.
>
> **Q1. About the literature review.**
>
> A1: We are very sorry that we didn't make clear the contribution so that reviewer might be misunderstood by our claims. Firstly, there is no doubt that [A] proposes message passing graph neural networks (MPGNN) which became the most common paradigm for graph neural networks like GCN, GAT, GraphSAGE, and etc. Moreover, they further incorporated some useful ways like "Virtual Edge"、"Master Node" and "set2set" to improve their GCN in **graph classification task**. Our model is actually implemented within the same message passing framework, which can be also considered as one of the MPGNNs. However,  our contribution is not proposing message passing framework, but indicating that **unifying**  the node feature and label propagation in both the training and testing phase can improve GNNs' performance in **semi-supervised node classification task**.
> We will update this statement in our literature review.
>
> [A] Gilmer, Justin, et al. "Neural Message Passing for Quantum Chemistry." ICML. 2017.

---

### Author Response · Authors · 2020-11-23
**Overall Response to Reviewers and Area Chairs**

We thank all the reviewers for their thorough and very helpful feedback. To summarize, all the reviewers agree on the effectiveness of our method, which proposes a novel framework to more effectively and explicitly utilize the label information by GNN in the semi-supervised scenario, achieving three SOTA results on Open Graph Benchmarks. However, there are still concerns about the motivation and experiments raise by the reviewers. And we have provided detailed responses to all your questions below with more experimental results and discussions, which will be added to the paper.


Please take a look and make a reply to let us know if you have any further concerns!

---

### Decision · Program_Chairs · 2021-01-07
**Final Decision**

**Decision:**

Reject

**Comment:**

This paper proposes a semi-supervised graph classification technique that unifies feature and label propagation techniques. The resulting algorithm is a simple extension that attains strong performance. Reviewers were divided on this submission. Some reviewers felt the proposed algorithm did not constitute a sufficient technical contribution given that it was a simple combination of existing techniques. I tend to agree with other reviewers that the simplicity is a benefit. However, despite the methods simplicity there was significant confusion about the details of the method and multiple reviewers flagged that the paper was difficult to read and understand. It further could benefit from additional discussion and some clarification/cleanup of the experimental results. Finally, multiple reviewers asked for better situating of the proposed method with respect to prior work. Given these concerns, I do not think the paper is ready for publication. I would recommend the reviewers do a thorough re-write of the paper to address these concerns and consider resubmitting.